# Subsurface warming derived by Argo floats during the 2022 Mediterranean marine heatwave

Annunziata Pirro[1], Riccardo Martellucci[1], Antonella Gallo[1], Elisabeth Kubin[1], Elena Mauri[1], Mélanie Juza[2], Giulio Notarstefano[1], Massimo Pacciaroni[1], Antonio Bussani[1], Milena Menna[1]

[1]National Institute of Oceanography and Applied Geophysics (OGS), Trieste, 34010, Italy
[2]Laboratory Balearic Islands Coastal Observing and Forecasting System (SOCIB), Palma, 07122, Spain

*Correspondence to*: Annunziata Pirro (apirro@ogs.it)

**Abstract.**

The Mediterranean marine heatwave (MHW) during the warm season (May-September) and the fall period (October-December) of 2022 is analyzed using Argo float in-situ observations in the upper 2000 m of depth. Five study regions (North Western Mediterranean, South Western Mediterranean, central Ionian Sea, Pelops Gyre and south Adriatic Pit) most affected by warming in different layers were selected and investigated. The primary goal is to provide insights into how the water column responds to the onset and progression of the MHW during the warming period, characterised by peak stratification and reduced vertical mixing. Additionally, this study aims to examine how the heat accumulated in the upper layers is redistributed to deep layers within regions with different dynamic characteristics through advection and/or mixing during the subsequent fall period.

Temperature anomaly profiles Ta (z) computed for each area and for both periods were divided into three categories based on vertical heat penetration: Category 1 (shallow, 0-150 m), Category 2 (intermediate, 150-700 m) and Category 3 (deep, > 700 m). During the warm season, Category 1 profiles had a temperature anomaly near zero or slightly negative in a thin layer between 50 m and 150 m depth, while warming was observed in the 0-50 m layer and below the middle layer. Profiles characterized by greater vertical heat penetration (categories 2 and 3) were mainly in mesoscale or sub basin structures and showed the largest positive temperature anomaly in the surface and intermediate layers. All profile categories showed a warming between 200 and 800 m depth. This increase is roughly split, with half attributed to the impact of the 2022 MHW, and the other half linked to the ongoing long-term trend in ocean temperatures. During the fall period and in the layer below 200 m depth, the shape of the Ta profiles are similar for all sectors with the exception of the south Adriatic Pit , which depict a $+0.5°$ C warming at 800 m depth.

The present work highlights the warming characteristics throughout the entire water column across different regions of the Mediterranean Sea, and seeks to connect the impacts of the warm season on the cold period with oceanic dynamic processes, such as dense water formation, upwelling or water column stratification. These regions are characterized by dynamic activities (e.g. dense water formation, upwelling), therefore, any variation in these ocean processes can influence the thermohaline circulation and, consequently, the climate system.

## Introduction

Marine heatwaves (MHWs) are extreme ocean temperature events occurring over extended periods of time (Hobday et al., 2016). Over the past decade the frequency of MHW events has increased by 50% (IPCC, 2023) as well as their duration and magnitude (Oliver et al., 2018). They can affect small areas of coastline or span multiple ocean areas across latitudes with significant impacts on ecosystems, coastal communities and economies (Wernberg et al., 2013; Garrabou et al., 2022; Dayan et al., 2023).

Since the beginning of the 21[st] century the particularly rapid warming trend of the Mediterranean Sea surface layer has been associated with a strong increase in MHWs events (Bensoussan et al., 2019, Ibrahim et al, 2021, Juza et al., 2022, Pastor and Khodayar, 2022, Dayan et al., 2023). Several studies, mainly confined at the surface, have addressed this topic covering different aspects of MHWs using satellite observations and model simulations. In particular, from basin to sub-regional scale, previous works analyze MHWs drivers and indicators, estimate the frequency, the duration and intensity of MHWs, evaluate their trend and assess the risk and the impacts on ecosystems (Darmaraki et al 2019, Galli et al., 2017, Garrabou et al., 2022, Juza et al., 2022, Dayan et al., 2023, Martinez et al., 2023, Marullo et al., 2023, Pastor and Khodayar, Simon et al., 2023). However, MHWs are not exclusively limited to the surface layer, but they can also propagate throughout the deeper layers of the water column (Darmaraki et al., 2019, Shijian et al., 2021, Scannell H.A., 2020, Juza et al., 2022). This can cause negative ecological consequences compromising the maintenance of the biodiversity, of the food and the regulation of air quality (Garrabou et al., 2022; Holbrook et al., 2020; Santora et al., 2020; Smale et al., 2019; Schaeffer and Roughan, 2017; Liquete et al., 2016; Martın-Lopez et al., 2016; Mills et al., 2013). A recent work in the Mediterranean Sea shows that although MHWs frequency is higher at the surface, their maximum intensity and duration is registered in the subsurface layers (Dayan et al., 2023). Moreover, in-situ data collected in the tropical western Pacific Ocean show that the maximum intensity of almost every MHW event is found in the subsurface layer, and many of the MHWs occurred even when no significant warming anomalies are detected at the surface (Shijian et al., 2021). Using satellite data, Marullo et al. (2023) defined the occurrence of the event in the Mediterranean Sea from May 2022 to spring 2023, with higher intensity in summer 2022 and in the band 0°-25° E. Starting from this result, the present work analyzes the subsurface properties of the 2022 MHW in the upper 2000 m depth using in-situ hydrographic Argo profiles (Product ref. no. 1, Table 1; Wong et al., 2020) collected during the period of highest

intensity (warm season, May-September) and in the period thereafter (cold season, October-December). Focusing on Marullo
et al. (2023) results  and on the availability of Argo float profiles, five study areas were selected for our analysis (Figure 1(b)).





| Product ref. no. | Product ID & type | Data access | Documentation |
|---|---|---|---|
| 1 | INSITU_MED_PHYBGCWAV_DISCRETE_MYNRT_013 _035; In-situ observations | EU Copernicus Marine Service Product, 2022a; | Quality Information Document (QUID): Wehde et al., (2022)<br><br>Product User Manual (PUM): In Situ TAC partners (2022) |
| 2 | MEDSEA_MULTIYEAR_PHY_006_004; numerical models | EU Copernicus Marine Service Product, 2022b; | Quality Information Document (QUID): Escudier al., (2022)<br><br> Product User Manual (PUM): Lecci et al., (2022) |

| | | | |
|---|---|---|---|
| 3 | SEALEVEL_EUR_PHY_L4_NRT_OBSERVATIONS_008 _060; satellite observations | EU Copernicus Marine Service Product, 2023; | Quality Information Document (QUID): Pujol al., (2023)<br><br>Product User Manual (PUM): Pujol., (2022a) |
| 4 | SEADATANET_MedSea_climatology_V2; climatology | SEADATANET Product; 2022 | Product Information Document (PIDoc): Simoncelli et al. (2020) |

**Table 1: Product data used to perform the analysis of the present work**

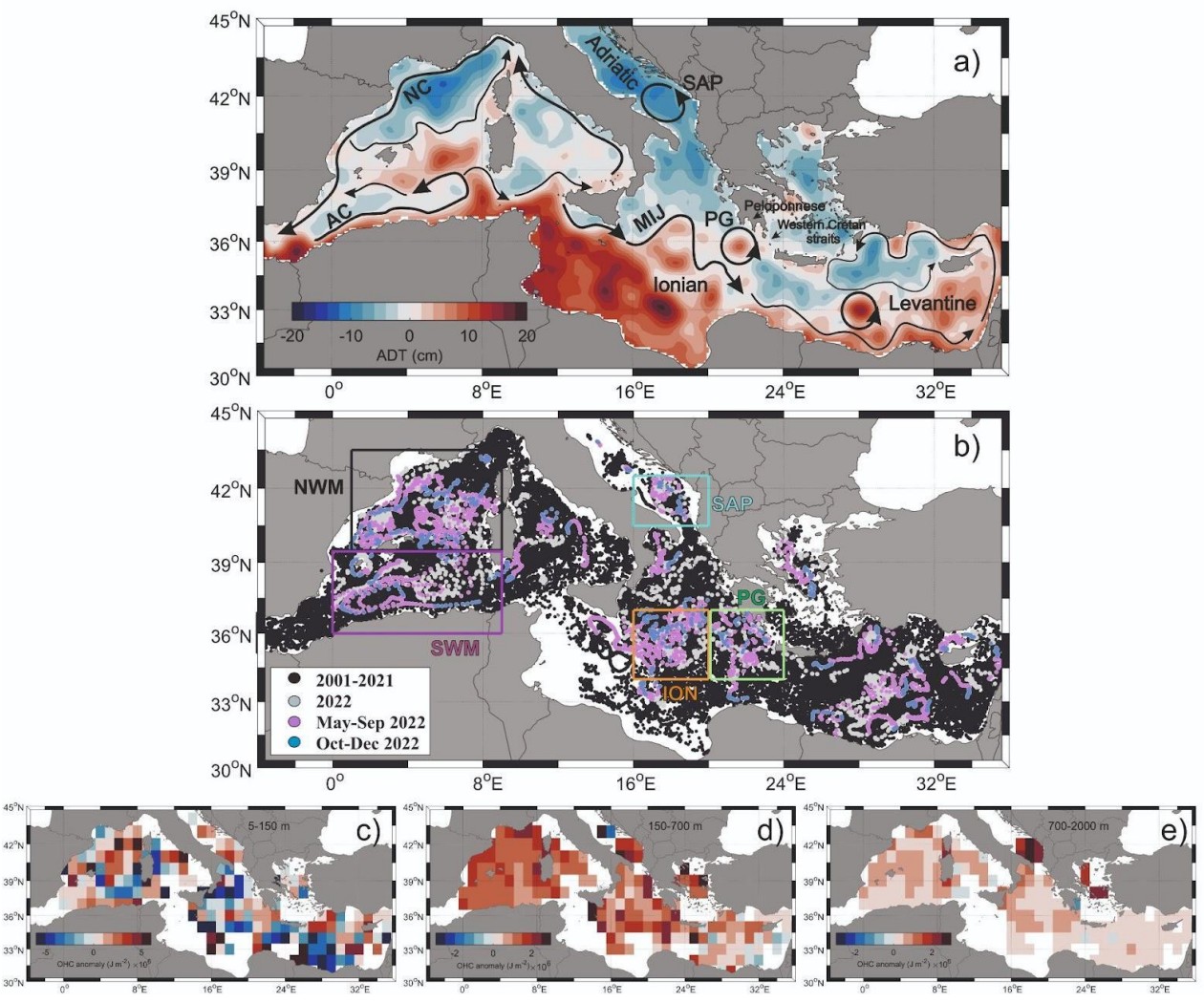


**Figure1: (a) Absolute Dynamic Topography (colours) averaged for the warm season (May-September 2022) along with schematic pathways (black arrows) of the Algerian Current (AC), Northern Current (NC), Mid-Ionian Jet (MIJ), South Adriatic Pit (SAP) and Pelops Gyre (PG). (b) Argo floats position for the whole Mediterranean Sea. Black, magenta, cyan, orange and green boxes indicate the North West Mediterranean (NWM, 39.5-43.5°N; 1-9°E), South West Mediterranean (SWM, 36-39.5°N; 0-9°E), South Adriatic Pit (SAP, 40.5-42.5°N; 16-20°E), Ionian (ION, 34-37°N; 13-20°E) and Pelops Gyre (PG, 34-37°N; 20-24°E) areas, respectively. (c-e) 2022 Ocean Heat Content (OHC) anomaly estimated every meter with respect to the 2001-2018 FLOAT climatology period from Argo floats profiles in different layers (c, 5-150m), (d, 150-700), (e, 700-2000).**

Based on the vertical heat penetration (MHW depth, see Methods section), the temperature profiles collected in May-September 2022 from each study area were divided into three categories (shallow, intermediate and deep penetration) and the median profile of temperature anomaly ($\tilde{T}_a$) was computed for each of them. Changes in the vertical temperature anomalies

were described and analyzed in relation to the ocean stratification, circulation and dynamics of each specific area. Lastly, this
study examines the properties of the water column during the fall period and speculates on its relationship with the dynamics
of the previous warm season's MHW. An estimation of the horizontal and vertical distribution of the Ocean Heat Content
(OHC) anomaly in 2022 was also performed in the whole Mediterranean Sea (Figures 1c-e).

## Methods

The vertical propagation of the 2022 MHW in the Mediterranean Sea was investigated using temperature data collected by
Argo floats in the period 2001-2022 (Figure 1(b)). These data were collected and made freely available by the International
Argo Program (which is part of the Global Ocean Observing System (Argo 2023)) and by the national program Argo Italy that
contributes to it (https://argo.ucsd.edu, last access 23 April 2023; https://www.ocean-ops.org, last access 23 April 2023).
A comprehensive characterization of the event over the whole Mediterranean Sea was performed starting from the OHC
analysis. The OHC, defined as the total amount of heat absorbed and stored by the ocean, can be considered as a good indicator
for assessing the Earth's energy imbalance (Von Schuckmann et al., 2016). A float derived OHC climatology ($OHC_{2001\text{-}2018}$)
for the period 2001-2018  was estimated in 1° x 1° bins and in different layers (0-150 m, 150-700 m, 700-2000 m) using the
method of Kubin et al., 2023. Subsequently, Argo temperature data collected in 2022 were averaged on the same grid of
$OHC_{2001\text{-}2018}$ to compute the 2022 OHC ($OHC_{2022}$). The $OHCA_{2022}$ was then calculated as the difference between $OHC_{2022}$ and
$OHC_{2001\text{-}2018}$ fields.
The five Mediterranean Sea regions most affected by surface warming  (Figure 1b) were selected using the results of Marullo
et al. (2023) and considering the availability of float data. In these regions we analyzed the vertical penetration of the 2022
MHW signal in the water column both during the warm and cold season. The regions selected are: the North Western
Mediterranean (NWM), the South Western Mediterranean (SWM), the Ionian (ION), the Southern Adriatic Pit (SAP) and the
Pelops Gyre (PG) sectors.
The Temperature anomaly $T_a$ at each depth z and for each profile was computed as:

$$T_a(z) = T(z) - \bar{T}(z), \qquad\qquad (1)$$

for each sector. T(z) is the 2022 temperature derived from Argo floats while $\underline{T}(z)$ is the climatological (1985-2018) averaged
temperature derived from the SeaDataCloud dataset (Product ref. no. 4, Table 1; SDC climatology). Specifically, the gridded
(0.125° x 0.125°) monthly climatological profiles were linearly interpolated in depth (every 10 m) and at the position of each
float profile. Moreover, to compare the 2022 MHW event with the averaged conditions estimated by floats in the selected
sectors, $T_a$ profiles were also computed for the whole float dataset in the period 2001-2018 (FLOAT climatology). It's important
to highlight that while this study utilizes the SDC climatology, the FLOAT climatology was utilized to facilitate a
straightforward comparison with the OHC findings from Kubin et al. (2023). The time window used for the present work
(May-September 2022) was chosen based on the latest European Space Agency specification
(https://www.esa.int/Applications/Observing_the_Earth/Mediterranean_Sea_hit_by_marine_heatwave, last access 18
February 2023) and on the estimations of Marullo et al. (2023) . These indicate that the 2022 MHW developed in the second
half of April in the northwest Mediterranean Sea and extended over the central Mediterranean into September. In this period,
$T_a$ profiles were quality controlled to remove any inconsistency (e.g. profiles with negative surface anomalies) and used to
estimate the vertical propagation of the MHW (or MHW depth), following the method of Elzahaby and Schaeffer 2019. For
each profile, the positive threshold depth (hereafter $Z_N$) is defined as the depth at which the first negative or 0 temperature
anomaly occurred:

$$Z_N = min \left( z(T_a \left( z \right) \leq 0) \right) , \qquad (2)$$

Knowing $Z_N$, the vertical cumulative temperature anomaly (CT$_a$) defined as:

$$CT_a(Z_N) = \sum_{z=0}^{Z_n} T_a(z) \, \Delta z , \qquad (3)$$

with $\Delta z$ = 10 m, was computed for each profile from the surface (z=0) to the positive threshold depth (z=ZN). To reduce the
effect of the insignificant warming at depths per water profile, we define the MHW depth as the depth where a fraction ($\varepsilon$=0.95)
of the cumulative $T_a$ is reached:

$$MHW \, depth = max \left( z \big( CT_a(z) \leq \varepsilon \cdot CT_a(Z_N) \big) \right) , \qquad (4)$$

Based on MHW depth values, $T_a$ profiles were then divided into three categories: Category 1 (shallow, 0-150 m), Category 2
(intermediate, 150-700 m) and Category 3 (deep, > 700 m). It's noteworthy that within the SAP area, float profiles categorized
as Category 2 and Category 3 consistently exhibit no negative temperature anomalies. However, they are classified into these
categories based on their respective depths, shallower or deeper than 700 meters. Additionally, despite the limited number of
profiles available in this region, they all fall within the cyclonic gyre. Hence, we are confident in considering them as
representative of the entire SAP region. The median profile ($\tilde{T}_a$) for each category was obtained by spatially averaging all the
available data in the different sectors in the warm period using 2022 and FLOAT climatology Argo data. Considering that the
2022 MHW extends until the spring of 2023, (Marullo et al., 2023), the median profiles $\tilde{T}_a$ for the fall period were also
examined to investigate the accumulation of the heat in the water column. The mean $T_a$ averaged in the surface, intermediate
and deep layers as well as other additional information (number of profiles, MHW depth, max $T_a$ and depth of max) are listed
in Table 2.

Lastly, the Brunt-Väisälä frequency squared ($N^2$) for the year 2022 and in the upper 150 m depth was computed using monthly averaged temperature and salinity Argo floats profiles for each sector in order to support the vertical heat penetration. The same procedure was adopted to calculate the $N^2$ anomaly with respect to FLOAT climatology.

| | | | Number of observations | MHW depth (m) | Temperature anomaly | | | averaged values | | |
|---|---|---|---|---|---|---|---|---|---|---|
| | | | | | Surface (10 m) | Max | Depth of max (m) | 0-150 m | 150-700 m | 700-2000 m |
| NWM | spring summer | C1 | 335 | 24,8 | 2,3 | 5,82 | 22,5 | *0.28 | 0,32 | 0,097 |
| | | C2 | 16 | 571,9 | 2,2 | 5,48 | 50 | 0,32 | 0,4 | NaN |
| | | C3 | 43 | 1457,9 | 2,92 | 5,58 | 19,5 | 0,8 | 0,36 | 0,1 |
| | | clim | - | - | - | - | - | 0,12 | 0,06 | 0,025 |
| | fall | fall | 306 | - | - | - | - | 0,66 | 0,33 | 0,11 |
| | | clim fall | - | - | - | - | - | 0,08 | 0,07 | 0,04 |
| SWM | spring summer | C1 | 159 | 25,6 | 2,13 | 5,79 | 22,5 | 0,19 | 0,33 | 0,088 |
| | | C2 | 5 | 630 | 1,83 | 5,46 | 24 | 0,43 | 0,3 | NaN |
| | | C3 | 27 | 1409,6 | 2,24 | 5,05 | 24,1 | 0,86 | 0,36 | 0,095 |
| | | clim | - | - | - | - | - | 0,028 | 0,059 | 0,028 |
| | fall | fall | 148 | - | - | - | - | 0,18 | 0,31 | 0,11 |
| | | clim fall | - | - | - | - | - | 0,1 | 0,05 | 0,02 |
| ION | spring summer | C1 | 105 | 22,8 | 1,34 | 4,58 | 22,2 | 0,03 | 0,27 | 0,12 |
| | | C2 | 5 | 644 | 2,18 | 2,87 | 18 | 0,58 | 0,35 | 0,54 |
| | | C3 | 3 | 1383,4 | 1,39 | 1,97 | 20 | 0,47 | 0,54 | 0,15 |
| | | clim | - | - | - | - | - | 0,071 | 0,091 | 0,057 |
| | fall | fall | 119 | - | - | - | - | -0,21 | 0,26 | 0,12 |
| | | clim fall | - | - | - | - | - | -0,06 | 0,07 | 0,05 |
| PG | spring summer | C1 | 50 | 37 | 1,34 | 3,82 | 41 | 0,15 | 0,32 | 0,03 |
| | | C2 | 15 | 553,4 | 0,95 | 6,15 | 47,3 | 0,97 | 0,34 | 0 |
| | | C3 | 20 | 1043,5 | 0,88 | 5,34 | 40 | 1,14 | 0,58 | 0,05 |
| | | clim | - | - | - | - | - | 0,3 | 0,15 | 0,02 |
| | fall | fall | 70 | - | - | - | - | -0,2 | 0,19 | -0,02 |
| | | clim fall | - | - | - | - | - | 0,27 | 0,13 | 0 |
| SAP | spring summer | C1 | 9 | 32,2 | 1,18 | 3 | 24,5 | 0,57 | 0,39 | 0,66 |
| | | C2 | 10 | 411 | 1,95 | 7,25 | 27 | 1,04 | 0,46 | NaN |
| | | C3 | 17 | 945,3 | 0,88 | 4,36 | 78,8 | 0,72 | 0,4 | 0,59 |
| | | clim | - | - | - | - | - | 0,3 | 0,21 | 0,21 |
| | fall | fall | 44 | - | - | - | - | 0,27 | 0,41 | 0,69 |
| | | clim fall | - | - | - | - | - | 0,29 | 0,2 | 0,16 |


**Table 2: Characteristics of the 2022 MHW in Category 1(C1), Category 2 (C2), Category 3 (C3): MHW depth, surface temperature**
**anomaly (Surface), maximum temperature anomaly (Max) and the depth where it occurs (Depth of max), mean temperature**
**anomaly for the surface (0-150 m), intermediate (150-700 m) and deep (700-2000 m) layers for each category and for the FLOAT**
**Climatology (clim).**

**Results and discussion**
In the surface layer, the $OHCA_{2022}$ displayed inhomogeneous warming patterns, with positive anomalies areas adjacent to
others with strong negative anomalies (Figure 1(c)). Largest positive anomalies were observed in the West Mediterranean, in
the South Adriatic, in the eastern Ionian and northern Levantine basin. In the intermediate and deep layers the warming was
more homogeneous and widespread (Figures 1(d), 1(e)) where the majority of bins showed positive values of the $OHCA_{2022}$
and specifically, the western and central Mediterranean areas along with the Aegean Sea showed a more pronounced warming
compared to the Levantine basin, which exhibits a slight cooling in some bins of the central and eastern sectors. It can be stated
that half of this warming in the intermediate and deep layers is due to the 2022 MHW while the other half to the long-term
warming of the ocean. This consideration stems from comparing the current $OHCA_{2022}$ with OHC trends defined by Kubin et
al. (2023). To perform this study, five regions (NWM, SWM, ION, SAP and PG; coloured boxes in Figure 1(b)) were selected.
This choice was motivated by the highest 2022 SST anomaly registered in the band 0 - 25° E (Marullo et al. 2023) and by the
availability of float data in both May-September and October-December 2022 periods. Figure 2 shows $\tilde{T}_a$ profiles for the warm
season of each sector, for each MHW depth category and for the FLOAT climatology.

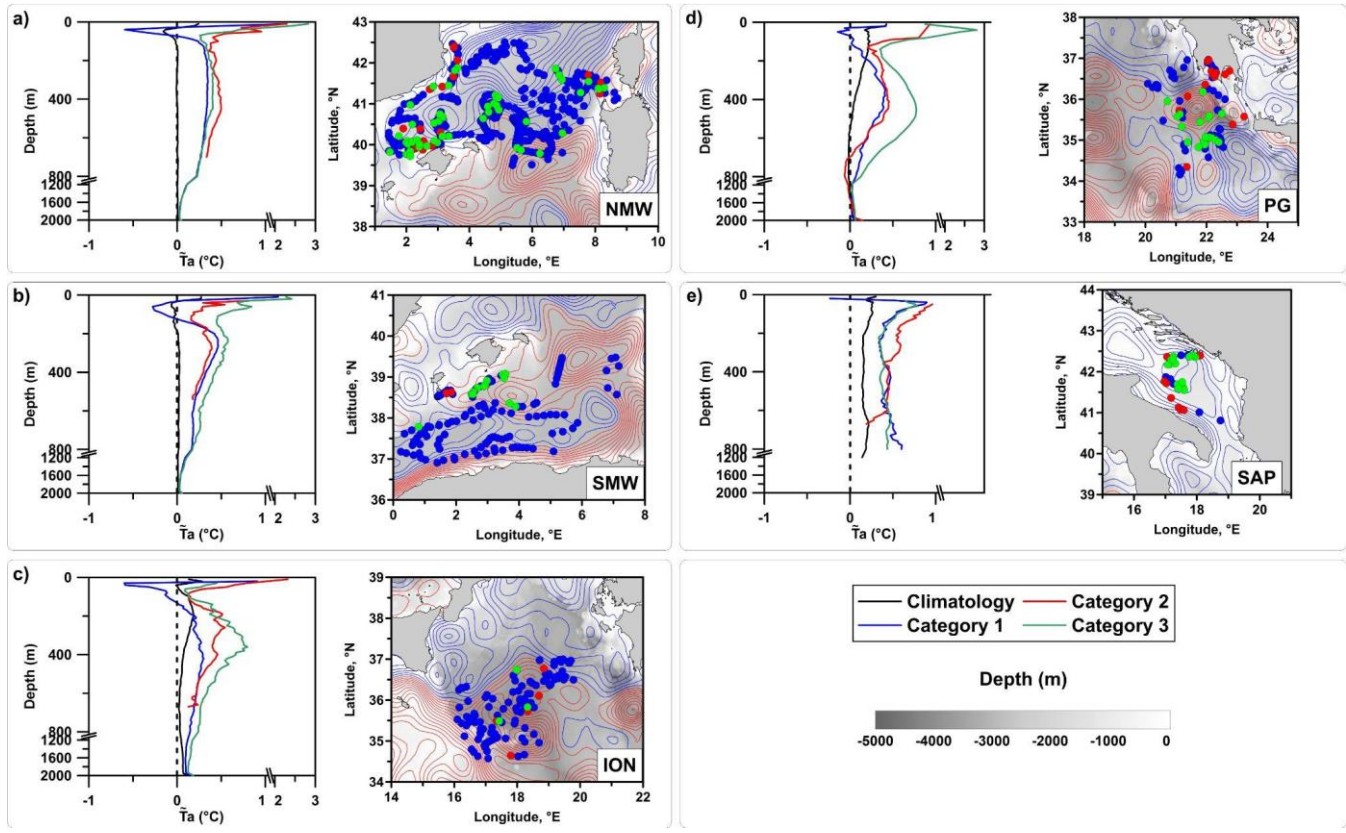


**Figure 2: (left panels) Median profiles of temperature anomaly computed for each sector (NWM, SWM, ION, PG, SAP) and for the 2022 warm season (May-September) using Argo floats data with respect to the 1985-2018 SDC climatology dataset. Black lines highlight the FLOAT climatology profiles while blue, red and green profiles indicate shallow (0-150 m), intermediate (150-700 m) and deep (> 700m) categories, respectively. (right panels) Positive and negative contours of the Absolute Dynamic Topography with 1 cm spacing are displayed by red and blue lines while the coloured dots are associated to the floats position of each category.**

In the NWM and SWM sectors the circulation is strongly influenced by the presence of two intense and permanent currents (Figure 1(a)): the south-westward Northern Current (Poulain et al., 2012; Escudier et al., 2021) and the eastward along-slope Algerian Current (which transports waters of Atlantic origin in the upper water column (Poulain et al., 2021)) in the NWM and in the SWM, respectively. Therefore, float profiles were mainly located along the boundary of cyclonic circuits as highlighted by the Absolute Dynamic Topography (Product ref. no. 3, Table 1; (Figures 2(a), 2(b)). In the ION sector, float profiles were mainly distributed in the anticyclonic meander of the Mid-Ionian Jet (Figure 2(c)), a strong meandering current that together with the Atlantic-Ionian Stream (AIS), transports Atlantic Water from the western to the eastern Mediterranean Sea (Poulain et al., 2012, 2013; Menna et al., 2019a; Figure 1(a)). Although the NWM, SWM and ION sectors have different oceanographic characteristics, they showed a similar response to the 2022 MHW (Figure 2(a-c)). Most $\tilde{T}_a$ profiles belong to Category 1 and the mean MHW depth falls into the 20-25 m layer (Table 2). Profiles, characterized by shallow MHW

penetration (blue lines in Figures 2(a-c)), showed decreasing warming in the first 50 m with the maximum $\tilde{T}_a$ close to the surface (22.2-22.5 m; Table2). The layer between 50 and 100 m depth showed a negative $\tilde{T}_a$ with maxima of -0.65° C, -0.2° C and -0.53° C at 50 m, 70 m and 40 m depth, in the NWM, SWM and ION sectors, respectively (Figures 2(a-c)). The median profiles derived from the FLOAT climatology (black lines in Figure 2(a-c)) do not exhibit this negative anomaly (or only to a very small extent), suggesting, therefore, a possible link between this behavior and the occurrence of the 2022 MHW. Below 100 m depth, the $\tilde{T}_a$ becomes positive again with mean values of ~ 0.3° C in the intermediate layer and values lower than 0.12° C in the deep layer. Profiles characterized by intermediate MHW penetration (red lines in Figures 2(a-c); MHW depth between 570 m and 650 m, Table 2) were located in coastal areas of the Western Mediterranean and in frontal zones in the ION sector, and showed positive $\tilde{T}_a$ throughout the water column, with values in the range of 0.3 - 0.6° C. Profiles, characterized by deep MHW penetration (green lines in Figures 2(a-c); MHW depth ~ 1400 m, Table 2), showed the largest $\tilde{T}_a$ in the surface layer in the two sectors of the West Mediterranean (> 0.8° C), while the ION sector depicted the largest anomalies in the intermediate layer (> 0.5° C). These results are consistent with the warming trend of the Western Mediterranean Sea over the last 15 years of 0.09±0.02 (0.03±0.01)° C·yr$^{-1}$ for surface (intermediate) waters (Kubin et al., 2023).

The PG is located on the eastern side of the northern Ionian Sea, southwest of the Peloponnese coast (Figure 1(a)). It is a sub-basin anticyclonic feature (diameter of ~120 km; Pinardi et al., 2015) which extends from the surface down to 800-1000 m depth (Malanotte-Rizzoli et al., 1997; Kovacevic et al., 2015) and it is forced by the Etesian winds (Ayoub et al., 1998; Mkhinini et al., 2014; Menna et al., 2021). In the late summer/fall the Etesian winds amplify their acceleration and the wind shear in the region of the western Cretan straits (Mkhinini et al., 2014) therefore, larger anticyclonic vorticities are observed during these months in the PG region (Menna et al., 2019a). In the sector PG, $\tilde{T}_a$ profiles for the three categories showed positive temperature anomalies in the first 800 m of the water column which coincides with the vertical extension of the gyre itself (Figure 2(d)). Profiles that fall into Category 1 showed decreasing warming in the first 70 m, anomaly values close to zero in the 70-150 m layer and increasing warming in the 150-400 m layer. The mean anomaly in the intermediate layer of Category 1 is 0.3°C (Table 2). Category 2 profiles were retrieved mainly in the coastal area near the Peloponnese while Category 3 profiles were found within the gyre area. Categories 2 and 3 showed strong warming in the surface layer (0.97° C and 1.14° C, respectively), a mean warming in the range of 0.3-0.6° C in the intermediate layer and no warming compared to the SDC climatology was observed in the deep layer (Table 2).

The SAP is one of the sites of open ocean convection in the Mediterranean Sea, characterized by a complex thermohaline circulation that influences the physical and biogeochemical properties of the dense waters formed in its interior and the strength of winter convection (Martellucci et al., 2024; Di Biagio et al., 2023; Menna et al., 2022 OSR6; Pirro et al., 2022). This sector showed positive temperature anomalies in all layers and in all categories (Figure 2(d)). Most profiles belong to Category 3 with a mean MHW depth of ~ 950 m and maximum $\tilde{T}_a$ at ~ 80 m depth. The largest mean warming was observed in the surface layer of each category (0.6-1.04° C) followed by the deep layer, which had an exceptional warming of ~ 0.6° C, and finally by the intermediate layer, with a mean warming of ~ 0.4° C (Table 2).

All five sectors showed a larger warming than the FLOAT climatology with a mean temperature increase in the 2022 warm
season between 0.2° C and 0.8° C in response to the MHW event (Table 2). Some differences in warming observed among the
sectors are related to their peculiar hydrological and dynamical characteristics. During the warm season, the surface layer of
the NWM and SWM sectors and partially of the ION sector, was characterized by both larger stratifications and stratification
anomalies compared to the FLOAT climatology (Figures 3(a), 3(b)). Strong stratification prevents vertical heat penetration
causing negative $\tilde{T}_a$ in the 50-100 m layer (Figure 2(a-c)). In the PG sector, warm season stratification anomaly was consistent
with climatology (Figure 3(b)), and vertical heat penetration was closely related to the gyre dynamics. In the SAP sector,
stratification during the warm period was lower than climatology suggesting an instability of the water column and therefore
the transport of the vertical heat to the deep layers. The median of all profiles available in 2022 warm season, when not
categorized, closely aligns with the median of profiles in category C1 (Figures 3(c), 3(d)). This condition arises because
category C1 consistently boasts the highest number of profiles across various sectors.

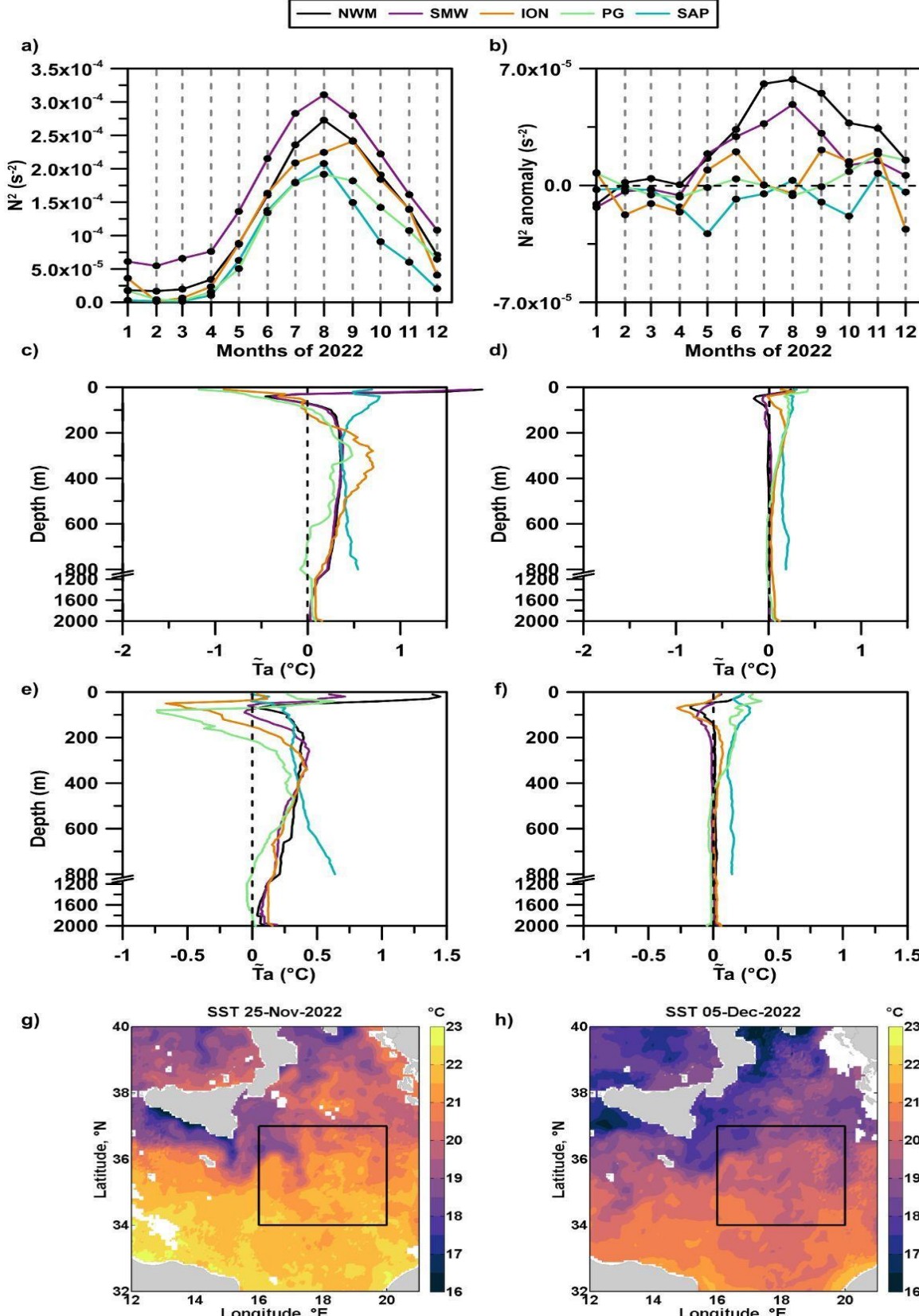


**Figure3: (a) Monthly averaged Brunt - Väisälä frequency squared (N2) computed in the surface layer (0-150 m) using 2022 Argo float data. (b) Monthly averaged Brunt Brunt—Väisälä frequency squared anomaly (N2 anomaly) computed in the surface layer with respect to the FLOAT climatology. (c) Median Temperature anomaly (°C) computed in the warm season (May - September) from Argo floats profiles in 2022 and (d) in 2001-2018 with respect to the SDC climatology. (e) Median Temperature anomaly (°C) computed in fall period (October - December) from Argo floats profiles in 2022 and (e) in 2001-2018 with respect to the SDC climatology). (g) Daily Sea Surface Temperature (°C) in the ION sector (black box) for late November and (h) early December 2022.**

Larger warming of the water column was observed in fall 2022 compared to the SDC climatology in all sectors, except for the surface layer of the ION and PG sectors (Figure 3e). The stronger warm season stratification observed in the NWM and SWM sectors (Figures 3(a), 3(b)) corresponds to enhanced vertical heat propagation in the surface and intermediate layers in fall 2022 (Figure 3(e), Table 2). Negative $\tilde{T}_a$ values in the surface layer of the ION sector were attributed to an upwelling event along the southern coast of Sicily between November and December 2022 as shown by the Sea Surface Temperature (Product ref. no. 2, Table 1; (Figures 3(g), 3(h)). The northern part of the Sicily Channel is an area of strong eddy kinetic energy (Poulain et al., 2012) influenced by Ekman transport and advection of waters from the western to the eastern Mediterranean (Molcard et al., 2002; Falcini et al., 2015; Schroeder et al., 2017; Menna et al., 2019b). The cold waters upwelled off the southern coast of Sicily in November 2022 (Figure 3(g)) were advected to the Ionian Sea through the Atlantic-Ionian Stream and the Mid-Ionian Jet pathways (Figure1(a)), and gradually cooling the waters in the ION sector (Figure 3(h)). The negative anomaly in the surface layer of the ION sector is not limited only to 2022 but is a permanent characteristic of the area related to the upwelling phenomena, as confirmed by the $\tilde{T}_a$ profile derived from the FLOAT climatology (orange line in Figure 3(f)) and by trends of the OHC anomaly estimated by Dayan et al. (2023) over the period 1987-2019. Negative $\tilde{T}_a$ values in the PG sector were imputable to the typical downwelling process of this region associated with the gyre dynamics. The downwelling contributed to the vertical propagation of the 2022 MHW, with a strong spring-summer warming in the first 800 m of the water column (Figure 2d), keeping the stratification values similar to the FLOAT climatology (no significant increases of $N^2$ anomaly was registered due to the 2022 heatwave; Figure 3(b)). In this way, fall cooling can penetrate deep into the water column causing, therefore, negative $\tilde{T}_a$ values in the surface layer (Figure 3(e).; Table 2).

In recent years, the SAP is experiencing a significant temperature increase in the deep layer (trend of $\sim 0.06°$ C·yr$^{-1}$ in the 2013-2020 period according to Kubin et al., 2023) and salinity in the surface and intermediate layers (Martellucci et al., 2024; Menna et al., 2022 OSR6; Mihanovich et al., 2021) with potential future effects on the whole thermohaline cell of the Eastern Mediterranean. It is of general understanding that convection sites contribute to the propagation of the MHWs signal from the surface to the subsurface interior of the water column (Dayan et al., 2023; Kubin et al., 2023) but specific analysis at the local scale are not yet available (Juza et al., 2022). Our results show a fair significant warming of the SAP in both spring-summer (Figures 2(e) and 3(c)) and fall (light blue line in Figure 3(f)) 2022 and a significant positive anomaly of FLOAT climatology compared to SDC one (black line in Figure 2(e) and light blue line in Figure 3(f)). In fall, largest $\tilde{T}_a$ in the SAP were observed in the deep layer ($\sim 0.69$ °C); Table 2, Figure 3(e)). Mean profiles derived from Float Climatology (black line in Figure 2(e)

and light blue line in Figure 3(f)) showed positive values compared to SDC one, confirming the warming trend throughout the water column over the past decade. Beyond the impact of the global warming of the Mediterranean Sea, the 2022 MHW leads to an additional heating in the SAP, which is transferred to the deeper layers favored by dynamical features of this area.

This study aims to characterize the 2022 MHW in the subsurface layers, and attempts to explain the mechanisms that drive the heat penetration to deep layers. However, further and more detailed investigations are needed to better support this last conclusion. We show that the effects of the 2022 MHW are felt in all layers of the Mediterranean Sea with vertical heat propagation extending from the surface to ~1500 m depth. In the surface layer, heat penetration and storage are related to the strength of the stratification and/or advection from adjacent regions. In contrast, the transport and the storage of heat in the intermediate and deep layers are closely linked to the dynamics of each area. These considerations are in line with the findings of Elzahaby et al. (2021) and Zhang et al. (2023), who noted that shallower MHWs are primarily influenced by surface air-sea fluxes, whereas deeper MHWs are predominantly driven by advection, manifesting distinct dynamics across various oceanic regions.

In the western Mediterranean and western Ionian Sea sectors, heat is mainly stored in the surface layer (shallow MHW depths and stronger stratification) so that this layer is significantly warmer than the climatology even during the following fall. Although deep MHW penetration in these regions is limited to coastal and frontal/eddies zones, it reaches the higher MHW depth estimated during the event. Sectors characterized by specific dynamics conditions (downwelling, convection) quickly distribute the heat in the water column even during the event. Intermediate layers exhibit comparable heating both during and after the MHW event, implying that heat can be stored there for extended periods and can be regarded as a long-term signal. The warming signal in the intermediate and deep layers could also be influenced by heat advection from adjacent basins however, we are aware that this topic needs to be studied in more detail in the future. In this context, the use of two climatologies and the cumulative anomaly threshold in the present analysis should have eliminated most of the signal associated with the ocean warming trend and advection therefore, the additional warming registered in spring-summer 2022 compared to the FLOAT climatology can be attributed to the effects of the 2022 MHW along the entire water column. Further studies are needed to investigate the effects that this warming may have on the physical and biological oceanic processes with implications on the thermohaline circulation of the entire Mediterranean Sea.

Competing interests. The contact author has declared that neither they nor their co-authors have any competing interests.

Author contributions. Conceptualization of the study was done by AP, MM and RM. AP and MM prepared the original manuscript. AP, MM, RM, EM, AG, GN, EK and MJ reviewed and edited the manuscript. AP, MM and RM created the methodology. AP, MM, RM and EK created the codes and performed the formal analysis. AP, MM, RM conducted the investigation. AG, AB and MP curated the data. EM was in charge of Argo-Italy infrastructure management and funding acquisition. All authors have read and agreed to the published version of the paper.

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
