# Peer review of "Subsurface warming derived by Argo floats during the 2022"

_State of the Planet, 2023_

## Author Comment (AC1)

*We thank both referees for the careful and insightful review of our report. We report our answers to the Specific and Technical comments provided by the reviewer*

**Review1**

**General Comments**

This study uses in-situ observations to identify the subsurface signal of the intense and long-lasting Mediterranean Sea marine heatwave of 2022. Beyond merely providing a description, the study links the propagation of MHWs by quantifying stratification and explaining the local dynamics (e.g., convection, upwelling). I find this study to provide convincing evidence of the vertical penetration of heat during the MHW, with new results on the depths influenced by these events. For example, an important distinction is made between the anomalous warming of 2022 and the long-term warming trend, which influence results on the deep profiles. While other referenced studies begin to study general characteristics of subsurface MHWs, this case study, performed by authors who display strong understanding of their study zone, is a welcome addition to the literature.

I agree with the publication of this manuscript after some minor corrections are made regarding clarity and communication. Generally, the figure quality could be improved. The manuscript is readable but would benefit from a grammar check (I provide technical corrections only for the abstract, for now).

**Specific Comments**

- Abstract: should mention the distinction between long-term warming signals and the 2022 anomalous warming. You mention the thermocline but do not refer to it or its depth in the main text. The last sentence is very vague. *Thanks, we will mention in the abstract the difference between the long-term signals and the 2022 warming signal. We have changed the last sentence according to the reviewer comment and will not mention the word thermocline in the abstract*

-

- Figures: inconsistency between fonts styles and size, map outlines and land colors. *Thank we will change it*

-

- Introduction: there is no mention of why it is important to look at subsurface penetration of MHWs, especially at the depths you study here. *The reviewer is right, we added it*

-

- Table 2 is blurry. *Thank you, we will make it more clear.*

-

- Line 39: "among others" is either lazy or unfair.

*We agree with the reviewer comment and added more references*

- Line 125: Could homogeneous warming be a sign that it is linked more to long-term warming than the 2022 event?

*The reviewer is correct that some of this warming could be due to the warming trend in the ocean. However, comparing the anomaly to the 2001-2018 period should reduce this effect. In addition, the OHC trend (per year) estimated for the Mediterranean Sea by Kubin et al. (2023) is an order of magnitude smaller than the anomaly in Figure 1d and 1e of this work or, in other words, the warming trend (e.g. in the intermediate layer) calculated in 15 years (see Kubin et al. 2023) is half of the 2022 warming presented in this work. Therefore, we will add a comment in the revised version of the manuscript explaining that half of the warming shown in the intermediate and deep layers is due to the 2022 MHW and half to the long-term ocean warming. .*

- In general, the paragraph separation could be improved to make the manuscript flow better. Line 140 is an example of an abrupt change. *Thank we will follow the suggestion*
- Line 235. This should be mentioned earlier in the results. As I was reading, I kept thinking "couldn't the deep warming just be long-term climate change signals"? It is best to tackle that doubt early. *Thank we will do it*
- Line 237: "A matter of fact" seems too confident to use generally. Possibly true in some places (e.g., SAP). *OK, we agree with the reviewer and we rephrased it*
- Line 240 is the same vague line found in the abstract. *OK, we changed it*

**Technical Corrections**

- Line 11: Argo float in-situ observations
*changed*

- Line 13: Pelops Gyre and southern Adriatic Sea,
*changed*

- Line 14: remove comma after "periods"
*changed*

 Line 14: T' – remove the apostrophe
*changed*

- Line 20: unclear why is SAP the exception.

*We are sorry for the confusion. The difference is that the SAP registered a larger warming compared to the warming depicted in the other sectors. We made it more clear in the text*

- **Review 2**

**overall quality**

- The manuscript studies the marine heatwave event that occurred in summer 2022 in the Mediterranean Sea using ARGO floats data. It first computes Ocean Heat Content anomaly (OHCA )for the surface, intermediate and deep layers for the year 2022 to determine regions that experienced strong warming. It then characterizes the vertical structure of the MHW event for each of those 5 regions using temperature anomaly profiles obtained from the ARGO floats during spring-summer 2022, to calculate the MHW depth that is then used to classify the MHW into 3 depth categories: surface, intermediate and deep. It concludes that different regions experienced MHWs with different vertical characteristics, with some regions showing MHWs constrained to the surface, and others MHWs propagating to intermediate and deeper depth. Authors state that the heat penetration can be explained by the dominant physical processes – whether stratification, advection, convection or upwelling – at play in each particular region.
The study provides insight into the vertical structure of MHWs by documenting MHWs beyond their surface signature, it also links the observed heat propagation to known regional ocean dynamics of the different regions however without support of an in-depth investigation for the hypotheses made. In the current state I would therefore not accept the manuscript for publication without substantial corrections; namely to carry out the evaluation of physical processes involved to support hypotheses, or to limit the scope of the study to the characterization of MHW with explanations of potential reasons for the observed signature.

*We thank the reviewer for carefully reading the report and for addressing below the major and minor comments. We will edit the report to accommodate the reviewer however, the reviewer should keep in mind that as stated at the end of the introduction, the report intends to characterize the 2022 MHW during the warm season and to show its effects during the fall period. We have linked the behavior of the water column during the fall period to some processes (e.g. dynamics of the area, stratification…) but, as the referee says, without fully supporting it. The reason for doing that, is that a report would not be enough to deeply study the physical processes occurring in all five areas and the proofs of these hypotheses are out of the report's scope. Therefore, we will make clear in the revised manuscript that these are potential reasons.*

**Specific comments**

- On the calculation of the 2022 OHCA, why is the OHCA calculated over the full 2022 year to determine study regions, when we know it is during the months of May to September that the MHW occurred? Why not just use the spring-summer period? If not, how can we ensure that the obtained $OHCA_{2022}$ value is not affected by a seasonal signal as a result of floats having heterogeneously sampled the region (e.g. only winter months)? Does an OHCA for summer months only lead to the same 5 regions?

*The reason for calculating the OHCA for the whole 2022, is because according to Marullo et al 2023, the strongest signal of the MHW starts in May and lasts until September but the signal is also observed until early 2023 (please see the figure of Marullo et al. below, as an answer to another your comment). Therefore, it is meaningless to reduce the computation of the OHC only to the May-Sept period. About the seasonal signal, we also have an answer for reviewer n.1 (please see above),*

*we are very confident that part of what we are looking is referring to the 2022 warming signal. In fact, when we compare the anomaly to the 2001-2018 period, this should reduce this effect. In addition, the OHC trend (per year) estimated for the Mediterranean Sea by Kubin et al. (2023) is an order of magnitude smaller than the anomaly in Figure 1d and 1e or, in other words, the warming trend (e.g. in the intermediate layer) calculated in 15 years (see Kubin et al. 2023) is half of the 2022 warming. Therefore, we can assess that half of the warming shown in the intermediate and deep layers is due to the 2022 MHW and half to the long-term ocean warming and we will add a comment in the revised version of the manuscript.*

- Also, the product used is from International Argo Program; why not use CORA product from the Copernicus marine service?

*From the Copernicus Marine Service we could have chosen many other products (beyond Argo floats or CORA), but since OGS is the Italian leader for the Argo Italy Program (for the deployment and the quality control), and since CORA product includes Argo floats, we prefer to conduct our analysis with the Argo floats profiles.*

- What is the justification for building an OHC climatology using argo data only? The presence of spatial gaps suggests limited data availability, how does this impact the calculation of OHC anomaly and associated uncertainties. Why not use a dataset containing more temperature data (like the Sea data Cloud dataset used later)?

*The reason for using Argo data to assess the OHC analysis is justified by several factors. First of all, the work of Kubin et al 2023 has already a float climatology (from 2001 to 2020) which is easily accessible. Secondly, the SDC climatology (from 1985 to 2018) is built on the Argo floats profiles and in-situ data, and considering that the in-situ data are limited compared to the floats profiles (on a ~ 20 years average) this would not strongly impact the mean value. As already stated in the previous comment, there are many products we can use to compute our analysis, but we think that using this climatology will not negatively affect the analysis. However in the revised manuscript we did not use anymore the OHC climatology from float to select the 5 study area, but our selection criteria is based on the work of Marullo et al (2023)*

- It would be informative to associate uncertainties to the OHCA, by knowing how many profiles are used for the calculation in each 1x1 degree gridcell when calculating the OHC climatology. Similarly for the OHC for spring-summer 2022. As additional plots may not be possible I would suggest to add an extra paragraph in the discussion.

*The number of float profiles in each 1x1 grid for the calculation of OHC are given in Figure 3 of Kubin et al. 2023 and here reported in the figure below, while the number of profiles considered for the 2022 warm period 2022 (and therefore for the OHC spring-summer 2022) are reported in Table 2 of the present report.*

[Figure]

FIGURE 3

Number of float profiles from 2001 to 2020 within 1°x1° grid boxes for **(A)** the surface layer (5-150 m); **(B)** the intermediate layer (150-700 m); **(C)** the deeper layer (700-2000 m) and **(D)** the deepest layer (1500-2000 m) measured by the Argo floats.

- Why the region centred on 28W 36N, south of **Türkiye** was not included in study as it seems affected by large anomalies (surface and middepth) and presents floats for both periods?

*The reason for not including it was because according to Marullo et al 2023, the high SST anomaly was registered from May to September 2022 in the Mediterranean Sea in the range 0-25E (please see the below figure from Marullo et al 2023). Additionally, the 28E-36N is a very small area to be considered and to make comparison with the other chosen areas. Also, if we consider a larger area, floats data are rarely available (see Figure 1b of the manuscript). We made clear in the text the criteria used to define the 5 study regions.*

[Figure]

*Figure from Marullo et al. 2023*

- Method for computing Ta. I am not a specialist but I wonder if there is a continuity between Argo data and SDC dataset so to ensure difference taken between float profile and mean SDC profile is meaningful?

*The reviewer is right and we apologize for not making it clear in the text. We are sure that the difference between the float profile and the SDC profile is meaningful because for each Argo profile available, we chose the closest 4 profiles from SDC (for the same month) and for the entire period 2001-2018, and we averaged them. We will make sure to specify it in the text*

- To evaluate the mean shape of MHW2022 vertical Ta profiles, a mean Ta profile is calculated using all floats for the season during year 2001-18. It is stated that this negative anomaly is not observed in the clim Ta profile (line 152), it would be useful to plot an uncertainty associated with the mean clim Ta profile, to get a better feeling of the spread of Ta with depth.

*In the lines 150-152 we simply state that (for Fig 2a,b,c) the black solid line (it represents the mean from 2001-2018 floats) compared to the zero dashed line (it represents the SDC2001-2018 climatology) does not display any negative values (it is always to the right of the SDC climatology). This suggests that the negative anomaly observed with the blue lines (in Fig 2a,b,c) and referring to the 2022 can be linked to the 2022 MHW therefore, we think that any other plot is not necessary to support this. However, to accommodate the reviewer request, we will make it more clear in the text. Regarding the associated uncertainty, please see below the STD calculated for the mean climatology profile. As we were expecting the STD is high in the surface layer since we are analyzing 20 years or so of data in 1x1 grid. Therefore, in the revised manuscript, we decided to use the Median and not the mean profiles, to have more consistent results (we also specified it in the comment below).*

[Figure]

- For each region a proportion of Ta profiles for each categories should be given
  *It is already reported in Table 2*

- In SAP region, how are Ta profiles categorized if (as suggested by mean profiles fig 2e) no negative anomalies are observed? Mean profiles for Category 1 and 3 seem identical, what makes them fall into different categories.

*SAP is a very special case. We have a smaller number of profiles than the other areas, but since all these profiles are limited to the same structure (topographic and wind-driven cyclonic gyre), they are representative of this area. Given the topography of the PIT, the depth varies greatly between the edges and the center (from a few hundred meters to 1200 m depth). The floats in categories C2 and C3 never show a negative temperature anomaly; they have only been included in these categories because they are shallower (C2) or deeper (C3) than 700 m.In category C1, there are 9 profiles (blue lines in Figure R1) that all show negative anomalies in the first 150 m (black profile in Figure R1), but the mean value does not represent them. For this reason, we decided to use the median profile (yellow in Figure R1) instead of the mean profile for all plots to represent the most recurring situation instead of the mean one. We will add a comment on this in the revised version of the manuscript.*

[Figure]

*Figure R1: Profiles of temperature anomalies in the SAP sector from the Argo float data in spring/summer compared to the 1985-2018 SDC climatology dataset. The black and yellow lines describe the mean and median of the profiles for each category. For category C1, a zoom to the first 200 m of the water column is shown (inset).*

- **purely technical corrections**
- General writing would benefit from being more precise, English could also be improved. *Thanks we will clear the text and improve the English*

Abstract
- Abstract could overall be improved,*We take into account the comment and will modify the abstract accordingly*
-
- line 13 to 17: the two sentences are counter intuitive and should be rewritten. First one mentions C1 has shallow heat penetration (0-150m) and the second sentence says C1 shows negative anomalies in that depth range (50-150) without mention of positive anomalies (at very near surface).Such info should be added.

*We thank the reviewer for this suggestion. We have improved the text in the revised version of the manuscript*

- line 18: study does not mention thermocline except for the abstract
  *The reviewer is right, we changed it*

- line 19-20: Unclear whether message conveyed is valid for both period studied or just summer (unlike the next sentence). Please clarify. *Thanks we clarified it*

- 'All sectors show similar warming', be more specific, in particular because a comparison is then made with SAP region. Unclear for reader what similar characteristics are observed and how this is different for SAP region.
  *The same comment was from Rew1. It was already taken into account.*

- Line 20: SAP use of acronym not yet defined. *Changed*
- Line 22: remove 's' to scenarios *changed*

Intro:
- Line 28: change to 'the frequency of MHW events has increased by 50%...' *Changed*
- Line 29: 'across latitude' strange use of term in this contect *We used it to indicate that MHW can affect areas at different latitude*
- Line 33: strong increase in MHWs, unclear number of MHWs?? *yes, number of events. we changed it*
- Line 34: 'facing' do you mean addressing? *yes, we changed in COVERING*
- Line 34: confine to and not in, *OK changed it*
- Line 40: propagate? I think the use of propagate is not appropriate, it suggests that warm anomalies travel from surface to deeper depth, when this may not be necessarily the only case with other mechanisms are also at play (lateral advection). This is also raised by the authors at the end of the paragraph with often no surface signature of mhws occurring at subsurface. Please rephrase. *we rephrased it*
- Line 43: registered, do you mean occurring? *yes*
- Line 45: 'were' detected *OK changed it*
- Line 71: calculated instead of computed, changes in and not of, *OK changed it*
- Line 72-75: a bit of a drafty sentence, please reword *OK changed it*

  Methods:

- Line 76-80: a brief *we changed it*

- Line 80: Argo product from the International Argo Program is not referenced in Table1 (or is it product 1?), *Yes, it is product1*

- Line 89: 'and better sampled by floats', but also sufficiently sampled by floats, *the word better has been changed with highly to indicate that the areas are chosen based on the OHC and high number of floats profiles.*
- Selected areas are referred to as region or sectors, please be consistent across manuscript in the term chosen, *Thanks we will do it*
- Line 90: The $T_a$ , please introduce before use, *changed*

- Line 93: what's the vertical resolution of SDC product? *Floats data are available every 10m, while the SDC have a higher resolution*
- Line 103: $Z_N$ not ZN, *changed*
- Definition of $Z_N$ line 104: if it's the temperature anomaly that is used in the expression it should be written as $T_a(z)$ and not $T(z)$, *thanks we changed it*
- Same for $CT_a$ *thanks we changed it*
- Line 108: by spatially *thanks we changed it*
- Line 110-111: please rephrase as not english*t hanks we changed it*
- Line 114-116: please explain why was $N^2$ calculated, *thank we did it*

Results:

- Line 130: specify in the text that profiles in fig2 correspond to those from the spring summer period. *thanks, we added it*
-
- Line 140-147 seems more part of a discussion aspect to me. *The report has a section results and discussion with no separation between them therefore, we can't move these lines to another section. In addition, results (e.g. floats position and therefore position of each category) are related and highly dependent to the "discussion" part (e.g. circulation of the Mediterranean). For these reasons we discuss these parts together.. We appreciate the reviewer's suggestion, but we can't accommodate it. The same answer is for the other comments below*
- Line 159-160: should be in discussion part, and needs elaborating. *please, see above comment*
- Line 161-166: should not be in results but in discussion *please, see above comment*

- Line 168: 'decreasing warming' specify the decrease is with depth, *thanks, changed*
- Line 174-176: same, description of area should not be in results *please, see above comment*
- Line 180: this is not true for Category 2 whose mean profile suggests a progressively decreasing Ta with depth.*Here, we refer to the mean Temp. for each category and for each sector which is clearly greater than the float climatology (see table2). Sorry for the confusion, we made it clear in the text*
- Line 180: sentence starting with 'All five sectors' is beginning of new paragraph *thanks, changed*
- Line 181: All five sectors showed a larger warming than the float clim: not the case for Cat1 in PG which is lower.*Here, we refer to the mean Temp of the 3 categories and for each layer as reported in Table 2. Sorry for the confusion, we made it clear in the text*
- Line 181-188 should be part of the discussion *please, see above comment*
- Caption fig 3: (e) Daily Sea Surface Temperature (°C) in the ION sector (black box) for late November and (f) early December 2022. *thanks, changed*
- Line 196: refer to relevant figure associated with statement, *thanks we added it*
- Line 200: attribution to an upwelling event also evidenced in N2 anomaly for the region in December *yes, it is correct*
-
- Note: mean Ta(z) are available for fall period, but not for summer period (available only for each category) which makes it difficult to visualize the seasonal change.

*We take into account the reviewer comment, and we will add a sub-plot to Fig3c showing the mean signal for both seasonal and for all the areas*

●
● References: Check formatting, presence of inconsistencies between citations (e.g. first names, …) *Thanks, we will do it*

---

## Author Response (AR1)

***Review n.2:***

*We extend our gratitude once again to both referees for their insightful review of our report. Based on their feedback and the editor's decision, the report underwent revision. Major alterations have been highlighted in green, although the majority of the report has been completely rewritten.*

---

## Author Response (AR2)

*We thank the reviewers and the referees for their comments. We report our answers below*

1) A second review is required.
As noted by reviewer 2, the region centered on 28E 36N shows strong anomalies (Fig 1c and d), perhaps in contradiction with the results of Marullo et al. (2023). A well-argued explanation should be provided.

*As already explained in the revised text and in the comments to the reviewers, our study areas were selected using the results of Marullo et al. (2023) and considering the availability of float data in summer (May-Sep 2022) and fall (Oct-Dec 2022). Even if we assume that the results from Marullo fail to detect all areas affected by the MHWs and want to compute the analysis for the area south of Turkey, this is not possible because the paucity of float data does not allow it. We find a total of 18 profiles in summer (10 in category C1, 0 in category C2 and 8 in category C3) and 3 in fall, all located in the pit southeast of Rhodes. The profiles are therefore few in number, do not cover the entire study period and are not representative of the overall dynamics of the area (see Figure R1 bottom right panel).*

2) A discussion on potential uncertainties related to the number of profiles has not been added.

*As already mentioned during the first round of review to reviewer 2 comments, the number of profiles used for the present study is reported in Table 2. Regarding the uncertainties, here we report the mean profile for each study area with the associated STD both, for the floats (May-Sept 2022) and the climatology data. The figure below shows that for each area investigated (NWM,SWM,SAP, ION,PG) the mean profile from floats (red line) is well within the climatology STD (blu area) therefore, it can be considered representative of the area investigated. Please, note that the float STD is represented by the dashed red lines. The same analysis was carried out in the sector south of Turkey, as proposed by reviewer 2, and it shows that the float data available in summer 2022 are not representative of the area (Figure R1 bottom right panel). This result definitely excludes the use of the sector south of Turkey in the present study.*

[Figure]

*Figure R1: Mean temperature profiles derived from the SDC climatology (blue) and the Argo float (red) with the associated standard deviations (blue shaded area for the SDC climatology and dashed red profiles for the Argo float) for the sectors used in this study and for the Turkish sector (TUR).*

3) Many of the relevant technical corrections suggested by reviewer 2 have not been taken into account.

*We take into account the comment, but have a different view. In the first round of review, we thoroughly addressed all comments of the Reviewer 2 in a point by point way, and edited the text accordingly. We apologize for any confusion caused by the absence of the tracked changes document in our initial submission. To clarify it, we have included the tracked document in this second round of revisions.*

4) A tracked version has to be provided to better assess the changes made.

*As stated above, we have provided it now.*

---

## Author Response (AR3)

*We thank the referee for reviewing our report and we changed the abstract according to the reviewer comment.*